# Haze Exposure Changes the Skin Fungal Community and Promotes the Growth of *Talaromyces* Strains

Dong Yan,[a] Min Li,[a] Wenhao Si,[a,c] Shijun Ni,[a] Xin Liu,[a] Yahan Chang,[a] Xiaochan Guo,[a] Jingjing Wang,[a] Jie Bai,[a] Yuanhang Chen,[a] Haoyue Jia,[a] Tao Zhang,[b] Minna Wu,[a] Xiangfeng Song,[d] Zhongwei Tian,[c] Liyan Yu[b]

aXinxiang Key Laboratory of Pathogenic Biology, Department of Pathogenic Biology, School of Basic Medical Sciences, Xinxiang Medical University, Xinxiang, Henan, China
bChina Pharmaceutical Culture Collection, Institute of Medicinal Biotechnology, Chinese Academy of Medical Sciences & Peking Union Medical College, Beijing, China
cDepartment of Dermatology, the First Affiliated Hospital of Xinxiang Medical University, Xinxiang, Henan, China
dDepartment of Immunology, School of Basic Medical Sciences, Xinxiang Medical University, Xinxiang, Henan, China

Dong Yan and Min Li contributed equally to this work. Author order was determined in order of increasing seniority.

**ABSTRACT** Haze pollution has been a public health issue. The skin microbiota, as a component of the first line of defense, is disturbed by environmental pollutants, which may have an impact on human health. A total of 74 skin samples from healthy students were collected during haze and nonhaze days in spring and winter. Significant differences of skin fungal community composition between haze and nonhaze days were observed in female and male samples in spring and male samples in winter based on unweighted UniFrac distance analysis. Phylogenetic diversity whole-tree indices and observed features were significantly increased during haze days in male samples in winter compared to nonhaze days, but no significant difference was observed in other groups. Dothideomycetes, Capnodiales, Mycosphaerellaceae, etc. were significantly enriched during nonhaze days, whereas Trichocomaceae, *Talaromyces*, and Pezizaceae were significantly enriched during haze days. Thus, five *Talaromyces* strains were isolated, and an *in vitro* culture experiment revealed that the growth of representative *Talaromyces* strains was increased at high concentrations of particulate matter, confirming the sequencing results. Furthermore, during haze days, the fungal community assembly was better fitted to a niche-based assembly model than during nonhaze days. *Talaromyces* enriched during haze days deviated from the neutral assembly process. Our findings provided a comprehensive characterization of the skin fungal community during haze and nonhaze days and elucidated novel insights into how haze exposure influences the skin fungal community.

**IMPORTANCE** Skin fungi play an important role in human health. Particulate matter (PM), the main haze pollutant, has been a public environmental threat. However, few studies have assessed the effects of air pollutants on skin fungi. Here, haze exposure influenced the diversity and composition of the skin fungal community. In an *in vitro* experiment, a high concentration of PM promoted the growth of *Talaromyces* strains. The fungal community assembly is better fitted to a niche-based assembly model during haze days. We anticipate that this study may provide new insights on the role of haze exposure disturbing the skin fungal community. It lays the groundwork for further clarifying the association between the changes of the skin fungal community and adverse health outcomes. Our study is the first to report the changes in the skin fungal community during haze and nonhaze days, which expands the understanding of the relationship between haze and skin fungi.

**KEYWORDS** skin fungi, haze, particulate matter, community composition, diversity

Address correspondence to Liyan Yu, yly@cpcc.ac.cn, Zhongwei Tian, zhonwt@xxmu.edu.cn, or Min Li, xxyxylimin@163.com.

The authors declare no conflict of interest.

Skin is the largest organ and home to millions of microorganisms in humans. The skin microbiota has important roles in the homeostasis as a component of the host defense (1–3). Fungi are less abundant than bacteria on the skin (4), but they play important roles in skin health (5, 6). Fungal communities are ignored by most of the studies of skin microbiota using high-throughput sequencing (7–10). Generally, the skin microbiota of healthy people is stable over time (11). However, it is influenced by a variety of factors, including age (12, 13), gender (14), and environmental variables such as climate (8), season (15), geography (16), hygiene practices (17), and urbanization (18, 19). Among the environmental variables, air pollutants are an important factor that disturbs the skin microbiota (20–24).

Haze exposure has been a threat to human health in developing industrial countries such as China (25–33). According to a report of the World Health Organization, approximately seven million people die each year as a result of diseases caused by air pollution exposure (34). Particulate matter (PM) is the primary pollutant in the air and is a complex combination of inorganic materials, organic molecules, and biological components (35). PM increases the risk of diseases and mortality in stroke (36), heart disease (37, 38), lung cancer (39, 40), respiratory diseases (41), and skin damage (42, 43). Considering that the skin is most directly in contact with PM pollutants, PM-bound microbes and chemicals can cause even higher degrees of harmful health effects. For example, long-term PM exposure causes skin inflammation, and even accelerates aging and wrinkle formation, affecting skin metabolism and destroying the skin barrier (43–45). PM is deposited on the skin and constitutes a part of the skin microbiota. The pollutants attached in PM may disturb the skin microbiota, affecting skin health.

Several studies have assessed the effects of pollutants on the skin microbiota (20–24). Exposure to $O_3$ and $NO_2$ significantly reduced the viability of skin bacteria (20, 22). Different polycyclic aromatic hydrocarbons and related xenobiotic chemicals have been shown to be degraded by skin bacteria (23, 24). Alterations in the composition and functional characteristics of the skin microbiota have been linked to chronic exposure levels of polycyclic aromatic hydrocarbon pollutants according to Leung et al. (21). However, to the best of our knowledge, the effect of the haze pollutants on skin fungi has never been evaluated. The goal of this study was to (i) determine whether haze exposure changes the skin fungal community composition and diversity, (ii) identify the genera associated with haze and confirm the interaction between the haze pollutants and the growth of representative strains of the genera *in vitro*, and (iii) reveal the assembly process of skin fungi during haze and nonhaze days.

## RESULTS

**The composition of the skin fungal community can be disturbed by haze.** A total of 74 skin samples were collected, including 19 samples during haze days (March 14 in spring), 19 samples during nonhaze days (April 27 in spring), 18 samples during haze days (November 29 in winter), and 18 samples during nonhaze days (December 30 in winter) in 2018. The levels of PM2.5 and PM10 and the air quality index (AQI) were recorded from the data of Xinxiang sites of the national program for recording urban air quality in China (https://air.cnemc.cn:18007) for 7 days before sample collection (Fig. 1 and Table S1 in the supplemental material). A total of 3,377,033 reads with 45,636 reads per sample on average were quality filtered, denoised, merged, filtered for chimeras, and clustered into 3,136 unique amplicon sequence variants (ASVs) (Table S2). The results of principal-coordinate analysis (PCoA) and analysis of similarities (ANOSIM) showed that the composition of the fungal community on the skin differed significantly between spring and winter samples (Fig. 2A, Fig. S1A), which suggested that the following analysis should be divided into spring and winter groups. According to the PCoA results, there was no significant difference in the skin fungal community composition between haze and nonhaze days (Fig. 2C, Fig. S1B). To further remove interference factors, we assessed other factors which can disturb the composition of the skin fungal community using canonical correspondence

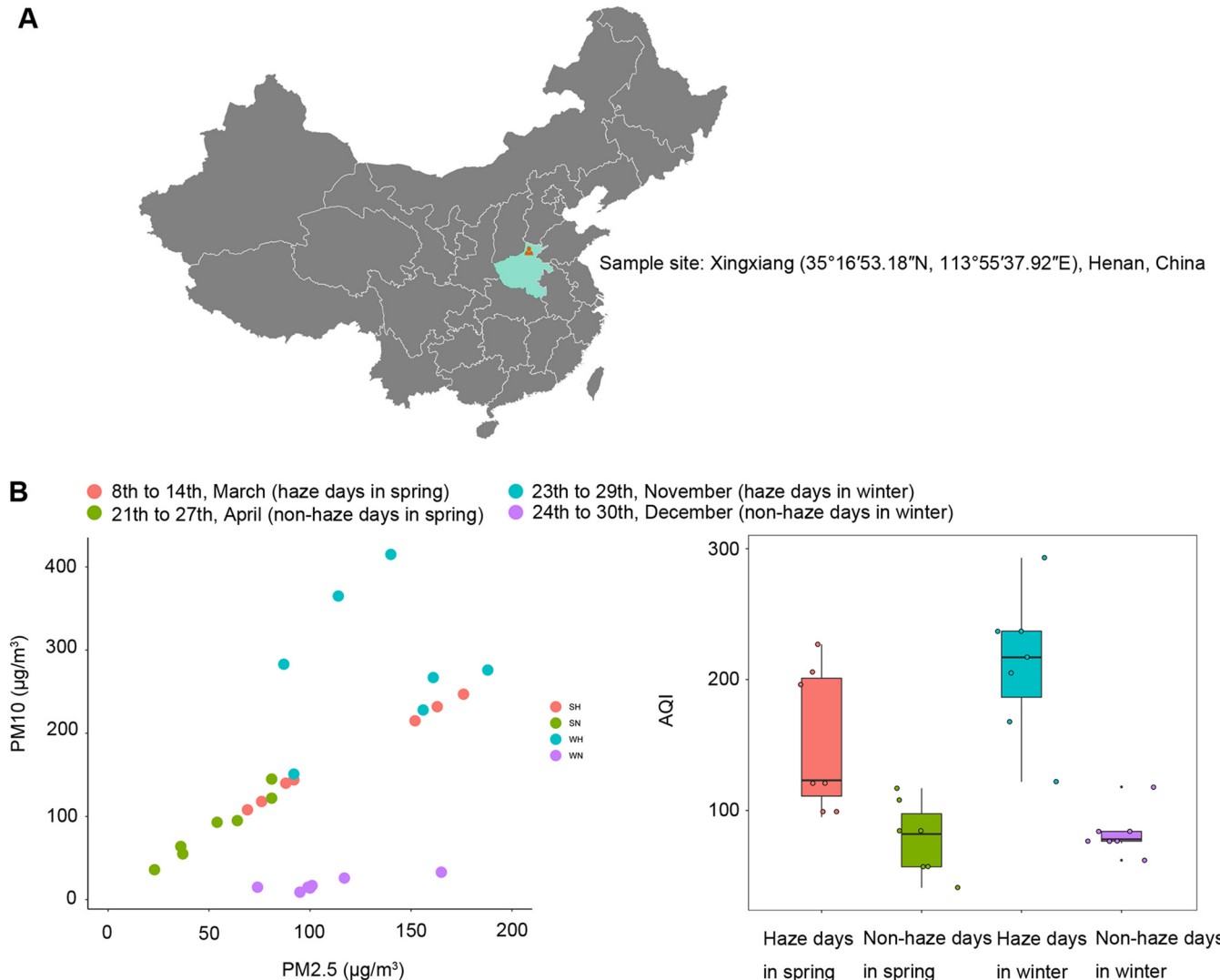

**FIG 1** The information of sample sites, date, and haze level. (A) Map showing the information of the sample site in Xinxiang, Henan, China. (B) The levels of PM2.5, PM10, and AQI on various sample days.

analysis (CCA). The CCA (Fig. 2B) and permutation tests (Tables S3 and S4) showed that gender had significant correlations with skin fungal community composition in spring ($R^2$ = 0.268, $P$ = 0.013) and winter ($R^2$ = 0.3623, $P$ = 0.001). Thus, we grouped the samples into female and male groups. Surprisingly, there were significant differences between haze and nonhaze days in both the female and male groups in spring and male samples in winter based on unweighted UniFrac distance analysis, confirming that the composition of the skin fungal community can be disturbed by haze (Fig. 2D and E and Fig. S1C and D).

**Alpha diversity and taxonomic composition of the skin fungal community.** The alpha diversity was assessed by the Shannon index, phylogenetic diversity (PD) whole-tree index, and observed features. The high-throughput sequencing captured the dominant phylotypes according to the rarefaction curves (Fig. S2). On average, 43 (from 9 to 92) and 48 (from 12 to 149) ASVs were observed during haze and nonhaze days, respectively, in spring samples. On average, 103 (from 40 to 182) and 80 (from 37 to 130) ASVs were observed during haze and nonhaze days, respectively, in winter samples (Table S5).

PD whole-tree ($P$ = 0.028) and observed features ($P$ = 0.045) indices were significantly increased in male samples during haze days compared to nonhaze days in winter, and Shannon indices also increased with no significant difference ($P$ = 0.42). These results indicated that haze increased the fungal diversities in male samples in winter.

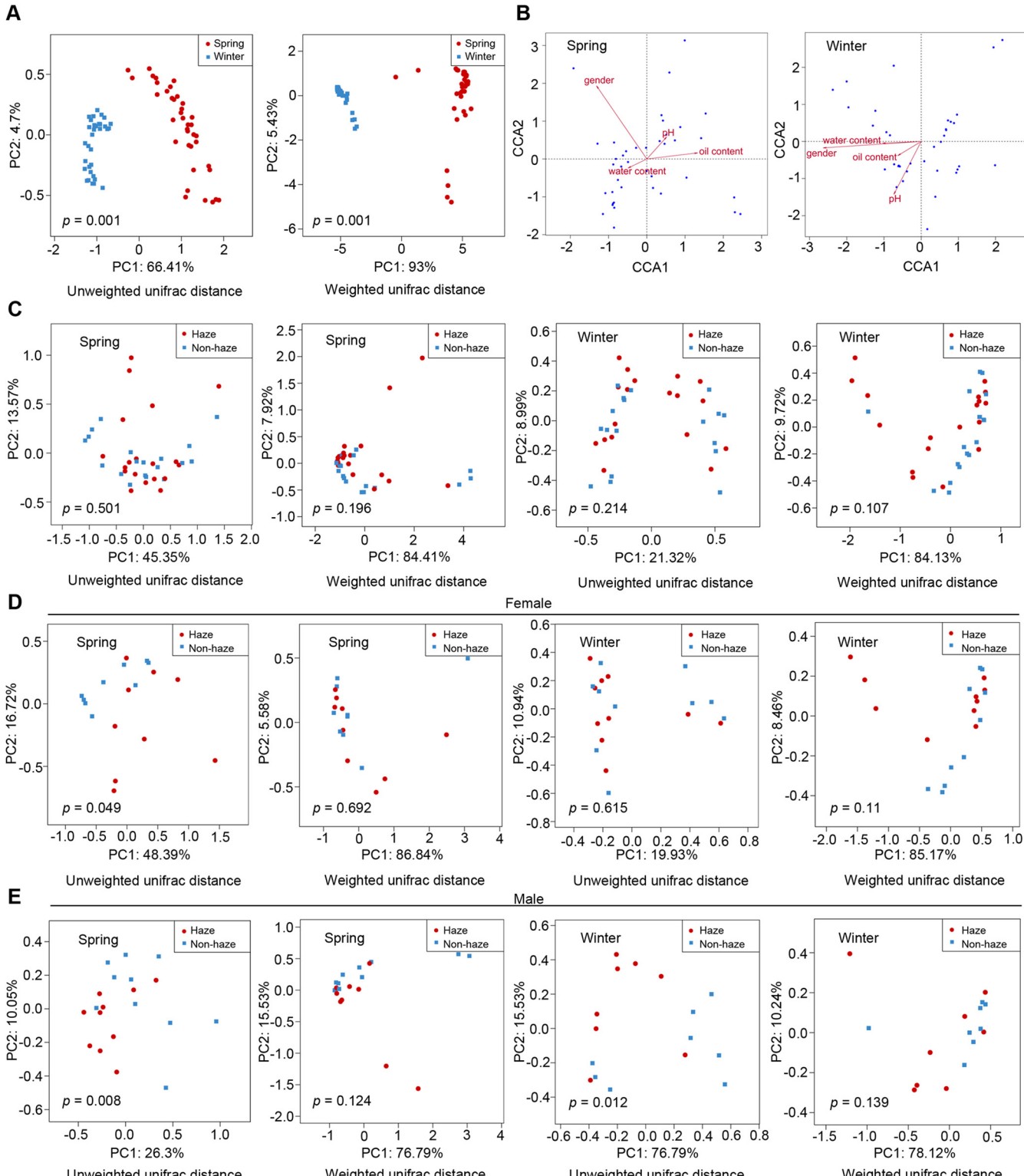

**FIG 2** Changes in skin fungal community composition during haze and nonhaze days. (A) Principal-coordinate analysis (PCoA) indicating changes in the skin fungal community between spring and winter based on unweighted and weighted UniFrac distance. (B) Canonical correspondence analysis (CCA) between skin fungal community and skin factors (gender, pH, oil content, and water content) in spring and winter. (C) PCoA indicating changes in the skin fungal community between haze and nonhaze days in spring and winter based on unweighted and weighted UniFrac distance. (D) PCoA indicating changes in the skin fungal community from females between haze and nonhaze days in spring and winter based on unweighted and weighted UniFrac distance. (E) PCoA indicating changes in the skin fungal community from males between haze and nonhaze days in spring and winter based on unweighted and weighted UniFrac distance.

There was no significant difference in other groups between haze and nonhaze days. In addition, compared to female samples, diversities were increased in male samples in both spring and winter (Fig. 3A).

In skin samples, 8 phyla, 87 orders, and 436 genera were discovered. The skin fungal community was dominated by Basidiomycota, followed by Ascomycota, Mucoromycota, Rozellomycota, Mortierellomycota, Chytridiomycota, Entomophthoromycota, and Glomeromycota (Fig. S3). The predominant order was Malasseziales (77.5%), which belongs to the Basidiomycota. Other dominant orders were Pleosporales (8.4%), Capnodiales (6.8%), Eurotiales (2.8%), and Hypocreales (1.4%), which belongs to the Ascomycota (Table S6). *Malassezia* (72.9%) was the most abundant genus in fungal community, followed by *Alternaria* (7.4%), *Cladosporium* (5.3%), Malasseziales unidentified (1.8%), and *Talaromyces* (2.3%) (Fig. 3B).

**Differences in skin fungal composition between haze and nonhaze days.** A heatmap diagram showed the distributions of dominant fungal genera (>0.02%) on the skin (Fig. 4A). For example, *Malassezia* was dominant in all samples. *Talaromyces* was increased during haze days; *Cladosporium* and *Mycosphaerella* were increased during nonhaze days in spring. In addition, some genera have different patterns depending on season or gender. For example, Malasseziales unidentified was more abundant in spring than in winter, while *Talaromyces* was less abundant. *Dothideales* unidentified, Sordariaceae unidentified, and *Zygosaccharomyces* were more abundant in female samples than male samples.

Linear discriminant analysis effect size (LEfSe) demonstrated that some fungal taxa differed significantly between haze and nonhaze samples (Fig. 4B). In spring, Dothideomycetes, Capnodiales, Mycosphaerellaceae, *Mycosphaerella*, *Cladosporium*, Cladosporiaceae, *Schizophyllum*, Schizophyllaceae, Agaricales, Ustilaginales, and Ustilaginomycetes were significantly increased during nonhaze days, Trichocomaceae and *Talaromyces* were significantly increased during haze days. In winter, Pezizaceae was significantly increased during haze days. Furthermore, four genera with significant differences were selected for further analysis using boxplots (Fig. 4C). *Talaromyces* was increased during haze days, while *Mycosphaerella* was increased during nonhaze days in spring. *Cladosporium* was increased during nonhaze days in spring, while it was decreased in winter. It is hard to indicated the differences, because the relative abundance of *Schizophyllum* was too low.

**High concentrations of PM promoted the growth of *Talaromyces* strains.** To further reveal the influence of haze exposure on the skin fungi, five *Talaromyces* strains were isolated (Table S7). The phylogenetic tree was constructed based on all *Talaromyces* sequences from high-throughput sequencing, isolation, and a type strain of *Talaromyces marneffei*, which clustered into four groups (Fig. 5A). The type strain of *Talaromyces marneffei* (a common pathogen) and XSF7 were found to be in the same group. One representative strain was selected from each group. XSF1, XSF7, XSF10, and XSF103 were selected to investigate the influence of PM on *Talaromyces* strains *in vitro*. The spore suspension of *Talaromyces* strains was cultured with PM at 0 mg/mL (control), 0.08 mg/mL (low-concentration [LC] group), 0.64 mg/mL (medium-concentration [MC] group), and 5.12 mg/mL (high-concentration [HC] group) for 5 days. The dry weights of *Talaromyces* strains (particularly XSF7, XSF10, and XSF103) were increased greatly in the HC group but not in the MC or LC groups (Fig. 5B), indicating that the growth of strains was promoted by high concentrations of PM. The results confirmed that haze exposure influenced the growth of several skin fungi.

**The skin fungal communities and taxa that increased during haze days deviated from the neutral assembly process.** Previous studies suggested that the Sloan neutral model can be used to predict the assembly process of the skin microbial community (19, 21). To further clarify the mechanism of changes in the skin fungal community, we assessed whether the skin fungi can be explained by the neutral model during haze and nonhaze days. The neutral model with a lower Akaike information criterion (AIC) score performed better than the binomial and Poisson models in predicting the assembly of the skin fungal community (Fig. 6A). The skin fungal community during haze days exhibited a larger tendency for niche-based assembly than during nonhaze

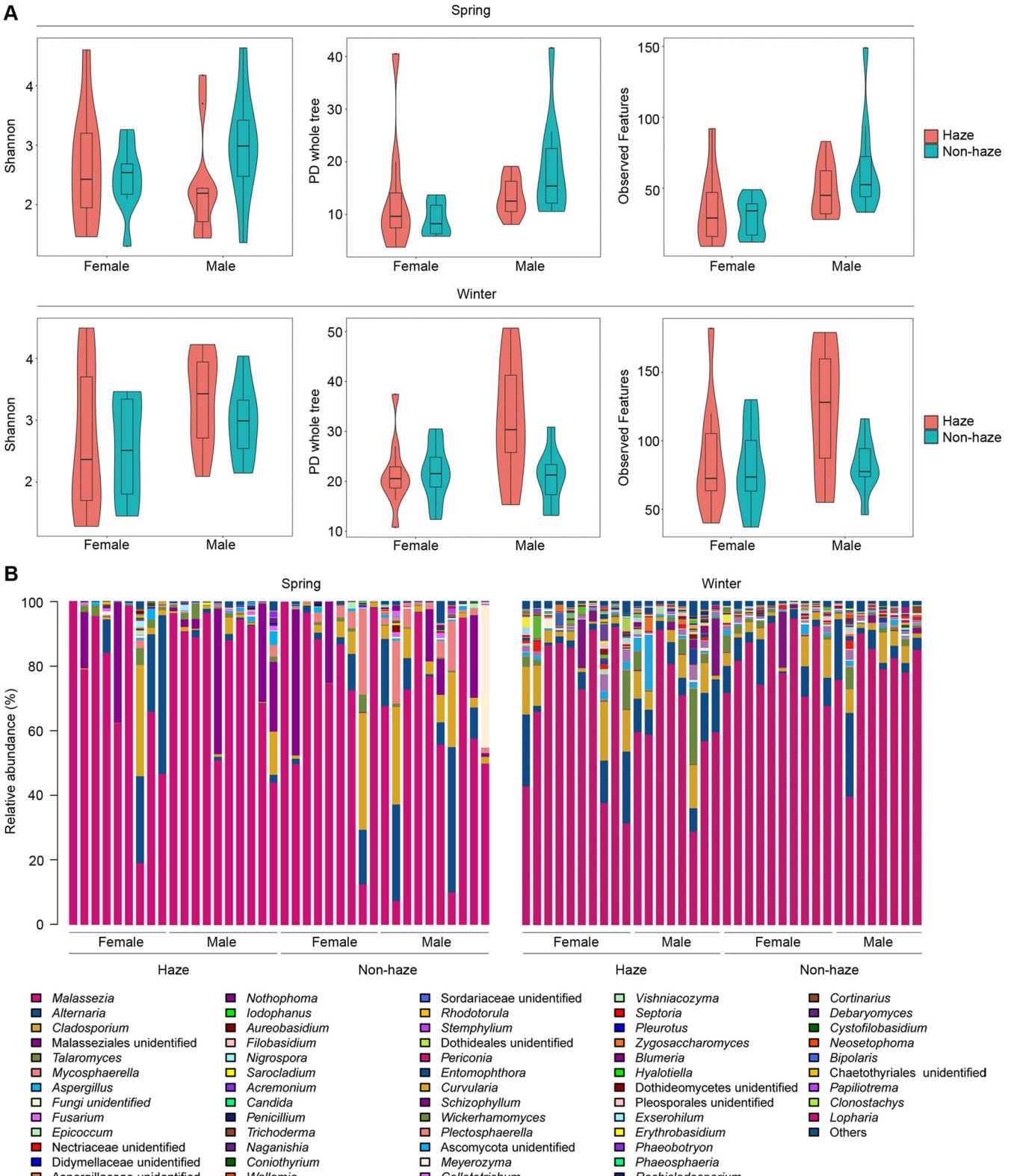

**FIG 3** Diversity and composition of the skin fungal community at the genus level. (A) Violin plot of skin fungal diversity based on Shannon, PD whole-tree, and observed features indices. (B) Bar plot indicating the relative abundance of dominant genera (>0.02%).

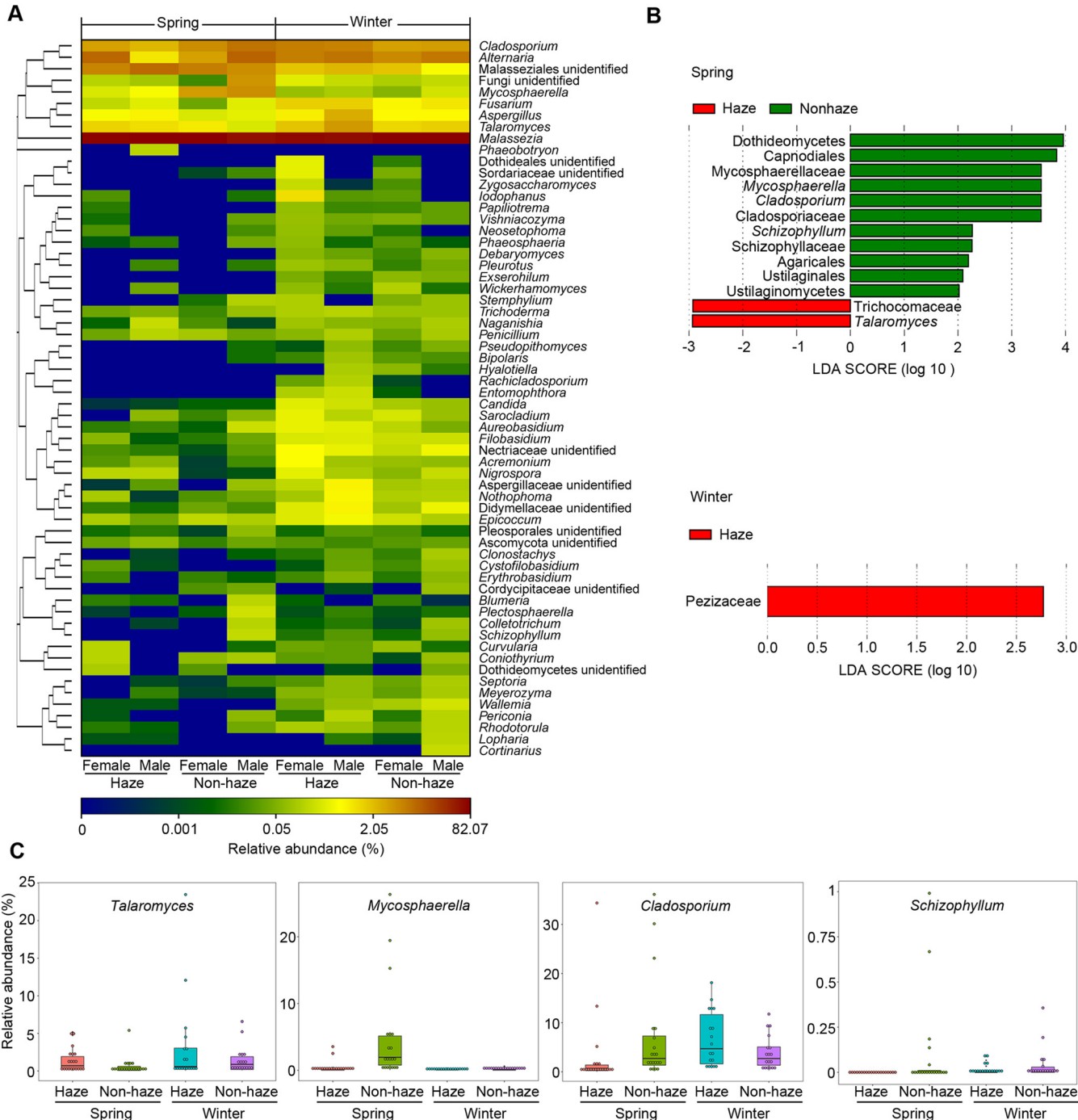

**FIG 4** The fungal genera with differences between haze and nonhaze days. (A) Heatmap analysis indicating relative abundances of dominant genera (>0.02%). (B) LEfSe analysis illustrating differentially abundant fungal genera among samples between haze and nonhaze days. (C) Variations in relative abundances of genera with significant differences (*Talaromyces*, *Mycosphaerella*, *Cladosporium*, *Schizophyllum*).

days in both spring and winter, which was demonstrated by a lower $R^2$ and migration rate (Nm) (Fig. 6B). In addition, compared to spring, the skin fungal community with the lower $R^2$ and Nm in winter indicated a larger tendency for niche-based assembly. Most of the fungal species were in the 95% confidence range of the neutral model (Fig. 6C). However, the fungal taxa significantly increased during haze days in winter, such as Trichocomaceae, *Talaromyces*, and Pezizaceae and had a higher percentage of species largely deviating from the neutral assembly process compared to the fungal community (Fig. 6C). The results predicted by the Sloan neutral model suggested that

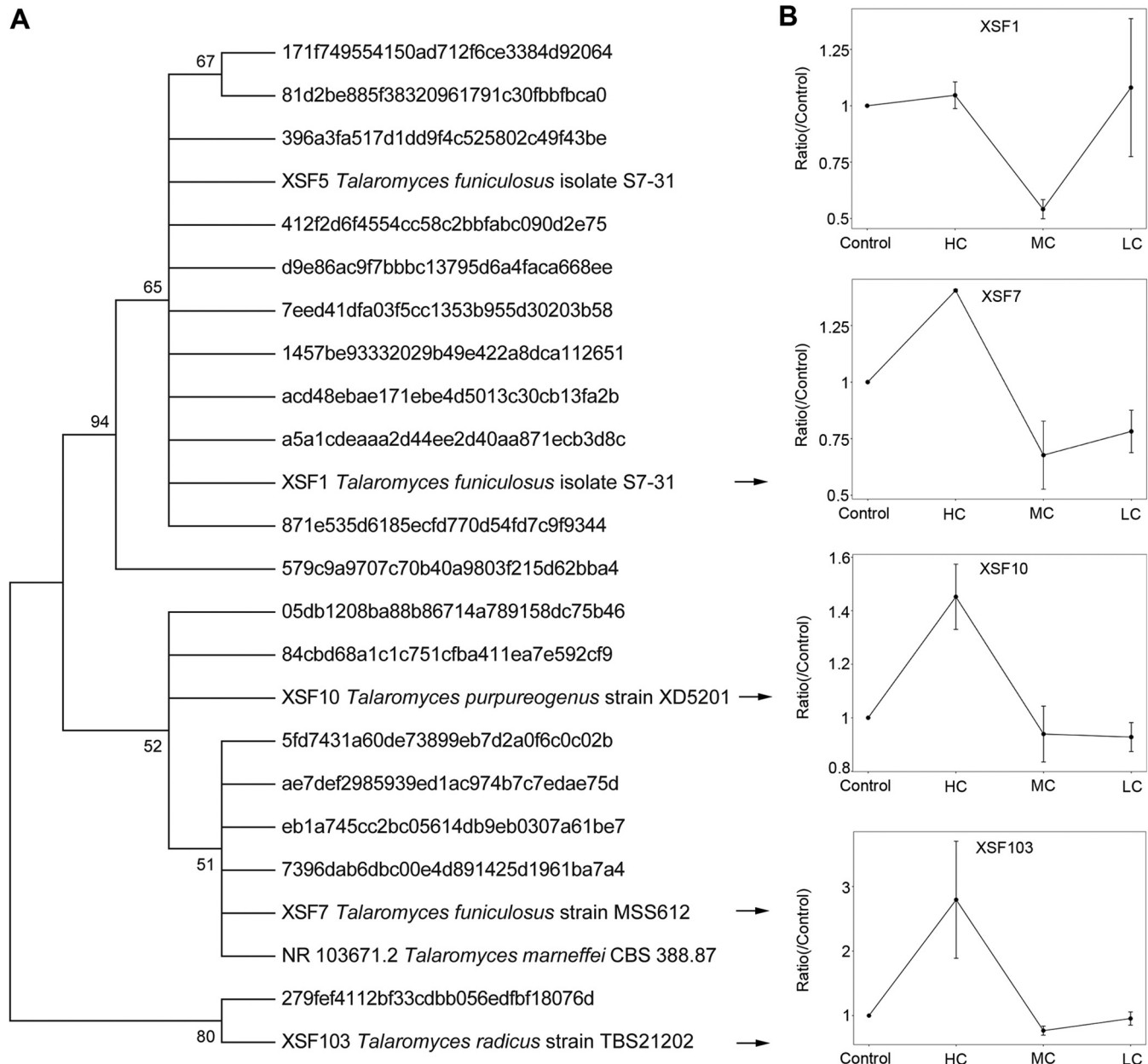

**FIG 5** High levels of PM pollutants promoted the growth of *Talaromyces* representative strains. (A) Phylogenetic tree of isolated *Talaromyces* strains based on ITS sequences using neighbor-joining methods (the closest strain is listed after each strain number). (B) High levels of PM pollutants promoted the growth of representative *Talaromyces* strains, especially in XSF7, XSF10, and XSF103. The *Talaromyces* strains were cultured with PM at 0 mg/mL (control), 0.08 mg/mL (LC group), 0.64 mg/mL (MC group), and 5.12 mg/mL (HC group).

the fungal community assembly was better fitted to a niche-based assembly model during haze days, and some genera with significant differences, such as *Talaromyces*, deviated from the neutral assembly process.

## DISCUSSION

Haze pollutants have been a universal public threat issue to human health, which can cause undesirable effects on the skin (21, 46). This study highlighted that haze exposure influenced the diversity and composition of the fungal community. *Talaromyces* strains were isolated and cultured with PM that we collected during heavy-haze days, confirming the influence of the haze pollutants on the fungi. Furthermore, the assembly process of the fungal community was associated with

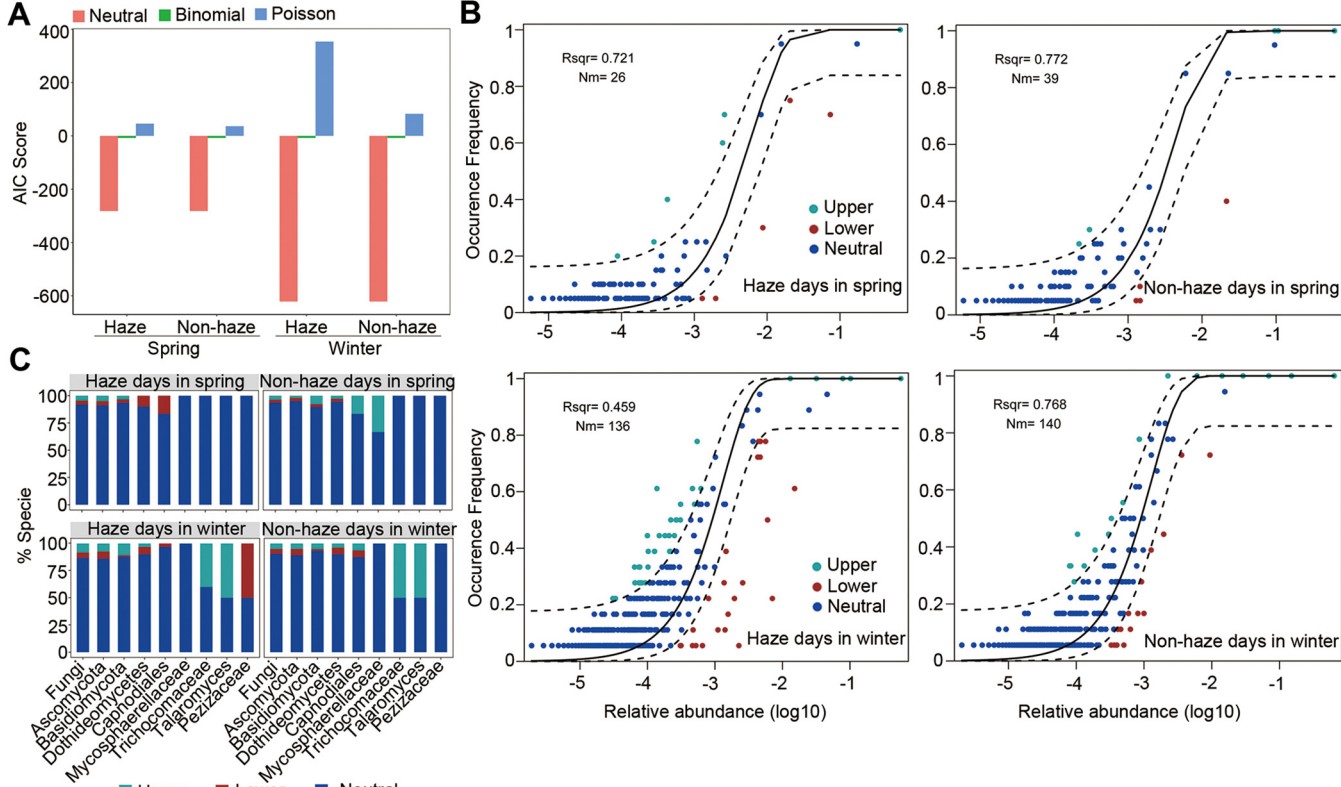

**FIG 6** Sloan neutral model predictions of the skin fungal community. (A) AIC score comparing the ability of the neutral, binomial, and Poisson models in explaining the skin fungal assembly process. (B) Sloan neutral model prediction in spring and winter during haze and non-haze days. Species are represented by data point and colored according to whether the taxon fit above (green), within (blue), or below (red) the 95% confidence interval (dotted lines). $R^2$ values (measurement of fit to neutral assembly process) and Nm values (estimated migration rate) are indicated for each prediction. (C) Proportion of species within each of the main PM pollutant-associated fungal taxa according to their Sloan neutral model prediction.

haze exposure. The exploration of the interplay between haze and skin fungi can elucidate new thoughts about how haze pollutants influence skin health by regulating the skin microbiota.

The skin fungal community can be influenced by various factors such as gender, body part, urbanization, season, pollutants, and so on (15, 18, 21, 47). In this study, significant differences in skin fungal composition between haze and nonhaze samples were observed after we removed interfering factors such as season and gender, indicating that the fungal community might be changed by haze, which is consistent with previous studies (21, 46). Polycyclic aromatic hydrocarbons (PAHs), a type of organic pollutants often associated with PM and known to induce skin diseases, were related to changes in the composition and functional capacity of the skin microbiota (21). In a German study, four bacterial strains from human skin degraded PAHs (23). He et al. observed that the skin microbiota of the forearm was almost halved after being exposed to $O_3$ (20). These studies confirmed the relationships between skin microbiota and air pollution.

Diversities in male samples from the winter group were higher during haze days than during nonhaze days in this study. Another study in China found that the increase of diversity was associated with PAH level (21), which was consistent with our study. Our previous study on airborne fungi associated with PM also indicated that Shannon indices were increased during haze days (31), implying that the increased diversity of the fungal community in the polluted air might be one of the reasons for the higher diversity of the skin fungal community during haze days. In addition, haze exposure increased the fungal diversity in the male group in winter, while no such pattern was observed in the female group. The variation may be

mainly related to the sebum level in the retroauricular crease, as adult males have higher levels of sebum than females (14).

The skin fungal community composition observed in this study is consistent with previous surveys of skin fungi (48, 49). For example, *Malassezia* is the predominant skin fungus in humans and is commonly observed on the skin of the face, scalp, and outer ears, which is rich in sebaceous glands (50). Despite having a clear correlation to skin diseases such as dandruff (51, 52), atopic dermatitis (53), and pityriasis (54), *Malassezia* species can also promote skin health like other skin-resident microbes by competing with the pathogen *Staphylococcus aureus*, secreting proinflammatory cytokines (55, 56), dissociating biofilms (57), and so on (58). *Alternaria* and *Cladosporium* were the dominant fungal genera in the skin samples. *Alternaria* and *Cladosporium* were the most prevalent genera in the air (59, 60) and dominated the airborne fungal community in China (31), which indicated that these fungal genera on the skin might come from the air. Moreover, *Alternaria* and *Cladosporium* species are considered risk factors to allergic disease due to their high allergenicity (60, 61), highlighting the high risk of allergic skin disease caused by the two genera.

With the accumulation of PM on the skin during haze days, the skin fungal composition can be changed. The taxa with significant differences were identified using LEfSe analysis. *Mycosphaerella* was significantly increased during nonhaze days in the spring group. It is always isolated from plants and contains one of the largest generic complexes of the pathogenic ascomycetes (62, 63). *Mycosphaerella* spp. can live in a range of habitats, such as symbionts, endophytes, hyperparasites, and plant pathogens, because it is saprobic (62, 64). Therefore, it is supposed that it lowered the rates of *Mycosphaerella* spp. released from plants during haze days with still air. *Talaromyces* was the only genus significantly increased during haze days. To further clarify the relationships between the haze and fungi, *Talaromyces* strains were isolated and cultured with PM. The results demonstrated that haze can promote the growth of *Talaromyces* at high concentrations. The genus *Talaromyces* has a worldwide distribution, and its species are food contaminants, mycotoxin producers, and human pathogens (65). For example, *T. marneffei* is a serious pathogen that can cause severe systemic infection by inhalation of conidia from the environment, which is the third-most-prevalent opportunistic infection in HIV patients in Southeast Asia and southern China (66). It can proliferate in macrophages, causing local infection in the skin and lungs as well as severe systemic disease (67). The sequence of the type strain *T. marneffei* CBS 388.87 from GenBank was clustered into one group with XSF7, indicating that PM may promote the growth of *T. marneffei* and increase the infection rate during haze exposure. In addition, *Talaromyces* species can produce abundant secondary metabolites with diverse bioactivities (68). For example, *Talaromyces pinophilus* has been reported to be able to degrade agricultural waste (69). Therefore, whether the significant changes of *Talaromyces* observed during haze exposure are associated with (or even cause) adverse health outcomes should be further studied.

The fungal community assembly was better suited to a niche-based assembly model during haze days. As demonstrated by Leung et al. (21), the scalp microbial community in the more heavily polluted city (Baoding, China) was better suited to the niche-based model, which is consistent with our observations. The retroauricular crease and the scalp are both enriched with sebaceous glands and have similar physiological characteristics, which explains the similar assembly process between our results and Leung et al.'s studies (21), while the cheek microbiota of individuals from the more lightly polluted city (Dalian, China) was better suited to a niche-based model (21), which suggested that the skin microbiota was influenced by the skin site and environment. According to the findings, microbial assembly is a dynamic process influenced by factors such as host physiology, environment, and season (11, 21). In addition, taxa with significant differences between haze and nonhaze days, such as *Talaromyces* and Pezizaceae, deviated from the neutral model, which indicated

that they had greater colonization potential (21, 70, 71). In particular, *Talaromyces* spp. enriched during haze days largely deviated from the neutral model prediction, suggesting that *Talaromyces* may be selected by haze pollutants and increased among the skin fungi community.

**Conclusion.** Our study first reported the changes of the skin fungal community during haze and nonhaze days. Haze exposure influenced the composition and diversity of the skin fungal community. Thus, the taxa with significant differences were observed, such as *Talaromyces*. Furthermore, the *in vitro* culture experiment revealed that the growth of representative *Talaromyces* strains was promoted at high concentrations of PM, confirming the high-throughput sequencing results. Finally, compared to nonhaze days, the fungal community assembly was better fitted to a niche-based assembly model during haze days. Several genera with significant differences, such as *Talaromyces*, deviated from the neutral assembly process. Our work provided a comprehensive characterization of the skin fungal community during haze and nonhaze days. It elucidates new insights on how haze exposure influences the skin fungal community, which provides the basis for further clarifying how the changes are associated with adverse health outcomes. We tried to clarify the interaction between haze and fungi at the strain level by the *in vitro* culture experiment, yet it is not enough. In the future, a combination of multiomics and the culture of haze pollutants and various fungal taxa will be required to confirm the relationships between haze, fungi, and skin health.

## MATERIALS AND METHODS

**Skin sample collection.** This research was approved by the ethics committee of Xinxiang Medical University. Healthy students aged 18 to 28 who resided on the campus of Xinxiang Medical University (35°16′53.18″N, 113°55′37.92″E) for at least 2 years were included in this study (Fig. 1A). The map was constructed using Pixel Map Generator (https://pixelmap.amcharts.com/). Exclusion criteria included a history of chronic disease, use of antimicrobials or topical steroids within 6 months prior to sampling, and bathing, shampooing, or moisturizing within 24 h before sampling.

The skin swab samples were collected from the skin of the retroauricular crease (behind the left and right ears). Sterile FLOQSwabs (Copan Flock Technologies, Italy) were dipped in a buffer with 0.15 M NaCl and 0.1% Tween 20. Regions of 5 cm by 5 cm were sampled. To access the site, the ear was folded forward with one hand to expose the crease. With the other hand, the shaft of the swab was held parallel to the surface of the skin and rubbed back and forth along the retroauricular crease approximately 50 times with firm pressure for 30 s. The head of the swab was inserted into the buffer-containing tube and cut from the handle aseptically. The unsampled swabs that had not been in contact with the skin were included as negative swab samples.

A total of 74 skin samples were collected on March 14 (haze days in the spring, 19 samples), April 27 (nonhaze days in the spring, 19 samples), November 29 (haze days in the winter, 18 samples), and December 30 (nonhaze days in the winter, 18 samples) in 2018. The pH, water content, and oil content of the skin were recorded on each sample day and are listed in Table S2. The AQI (72) was used to determine the haze levels. A day with an AQI of >100 was defined as a haze day, and one with an AQI of <100 was defined as a nonhaze day. The levels of PM2.5, PM10, and AQI were recorded from the data of Xinxiang sites of the national the national program for recording urban air quality in China (https://air.cnemc.cn:18007) for 7 days before sample collection (Fig. 1 and Table S1).

**DNA extraction and PCR.** Three blank swabs were used to generate genomic DNA as a negative control. The swab tubes were vortexed for 10 min before DNA extraction. The DNeasy PowerSoil kit (Qiagen, Germany) was used to extract genomic DNA from all samples, including negative swab samples. The fungal internal transcribed spacer 1 (ITS1) region of the rRNA gene was amplified by PCR (95°C for 3 min, followed by 35 cycles at 95°C for 30 s, 55°C for 40 s, and 72°C for 45 s, and a final extension at 72°C for 10 min) using primers ITS5 and ITS2 (73). *Ex Taq* DNA polymerase (TaKaRa, Japan) was used to conduct the PCR amplification. PCRs were performed in triplicate in 20-$\mu$L mixtures containing 4 $\mu$L of 5× *Ex Taq* buffer, 2 $\mu$L of 2.5 mM deoxynucleoside triphosphate (dNTP), 0.8 $\mu$L of each primer (5 $\mu$M), 0.4 $\mu$L of *Ex Taq* DNA polymerase, and 10 ng of template DNA.

**Illumina MiSeq sequencing.** Amplicons were purified using the AxyPrep DNA gel extraction kit (Axygen, USA) according to the manufacturer's instructions and quantified using QuantiFluor-ST (Promega, USA). The purified amplicons were sequenced (2 × 300 bp) on an Illumina MiSeq platform.

**Sequencing data processing and analyses.** Quality-filtering and denoising of raw reads and the taxonomy of ITS1 sequences against the UNITE database (version 04.02.2020, http://unite.ut.ee) (74) were conducted using QIIME 2 (75). PCoA and the analyses of diversities with the Shannon index, the PD whole-tree index, and observed features were performed using QIIME 2 (75). The PD whole-tree index represents the diversity based on the phylogenetic tree (76), and observed feature represent the richness of the fungal community. R software (version 4.0.2) was used to create the scatter, box, line, violin, bar, heatmap, and graphs, and CCA (77). LEfSe was used to identify the taxa with significant differences

between haze and nonhaze days (78). Taxa with a log linear discriminant analysis (LDA) score of >2.5 were considered and plotted.

**Cultivation, isolation, and identification of *Talaromyces* strains.** The buffer-containing swab sample was diluted six times and vortexed for 10 min before cultivation. Then, 100 $\mu$L of solution was spread-plated onto Sabouraud dextrose agar (SDA) medium (Oxoid, UK) and potato dextrose agar medium (Oxoid). To inhibit bacterial growth, the media were both supplemented with strepto-mycin sulfate and tetracycline (0.05 mg/mL). After incubation at 28°C for 5 days, the phenotypically distinct colonies were transferred onto corresponding agar slant. Chelex-100 was used to extract the genomic DNA of the strains. The fungal ITS regions were amplified by PCR (95°C for 5 min, followed by 35 cycles at 95°C for 30 s, 55°C for 45 s, and 72°C for 45 s, with a final extension at 72°C for 10 min) using primers ITS1 and ITS4 (79). PCR products were purified and Sanger sequenced at Sangon Biotech (Shanghai, China). The 60 bp of the beginning ITS reads were cut off and cut into 410 bp with BioEdit software to ensure the accuracy of the sequences. To identify fungal species, the top hit from BLAST analysis and GenBank was employed. Five *Talaromyces* strains were obtained and frozen at −80°C in 20% glycerol. The detailed information of the strains is listed in Table S7. The sequences of the type strain *Talaromyces marneffei* CBS 388.87 was downloaded from NCBI GenBank (accession number NR_103671) and aligned using MAFFT online (80). The phylogenetic trees were developed using MEGA software (version 7) based on the neighbor-joining method (81). Four representative strains were selected for further study.

**Collection and processing of PM.** Sampling was conducted using the LB-120F PM sampler (Lubo, China). PM was drawn at 100 L/min and collected on 80-mm glass fiber aerosol collection filters during haze days. The sampling filter was replaced with another one after 2 h, and 20 filters were collected. The filters were cut in half, immersed in 60 mL of deionized water, and sonicated for 30 min after sampling. The filters were immersed in another 60 mL of deionized water for another 30 min. The resulting mixed solution was centrifuged at 1,300 × $g$ for 10 min. The supernatants were sterilized through a 0.22-$\mu$m fil-ter, while the precipitations of PM were autoclaved. The two parts were mixed and vacuum freeze-dried overnight before being stored at −80°C.

**Culture of the representative *Talaromyces* strains and PM.** The representative *Talaromyces* strains were inoculated onto the SDA slant and cultured for 7 days at 28°C. To prepare the spore suspension, the spores were scraped using a sterile pipette and put into 1 mL of Sabouraud dex-trose broth (SDB) in the tube. The suspension was vortexed for 2 min and allowed to stand for 3 min before being transferred to another tube without big particles floating or sinking. The optical den-sity at 570 nm (OD$_{570}$) of the suspension was adjusted to 0.2. Then, 10 $\mu$L of spore suspension was inoculated in 2 mL of SDB and cultured with PM at 0 mg/mL (control), 0.08 mg/mL (LC group), 0.64 mg/mL (MC group), or 5.12 mg/mL (HC group) for 5 days. Three replicates of each strain were conducted in each group. The dry weights were measured, and the ratios of HC, MC, or LC to control were used to estimate the influence of PM pollutants on the *Talaromyces* representative strains.

**Prediction by the Sloan neutral model.** To explore the mechanism of changes in the skin fungal community, Sloan neutral model prediction (82) was conducted as described previously (19, 21). The migration rate (Nm) is an indication of dispersal limitation, and a higher $R^2$ value is an indication of fit-ting to the neutral model. Based on AIC scores, the three common models (neutral, binomial, and Poisson model) were compared with each other.

**Ethics approval and consent to participate.** The research protocol for taking samples from volun-teers was approved by the ethics committee of Xinxiang Medical University.

**Data and materials availability.** The raw sequencing data have been deposited at the NCBI Sequence Read Archive (SRA) database under accession number PRJNA796697.

## SUPPLEMENTAL MATERIAL

Supplemental material is available online only.
**SUPPLEMENTAL FILE 1**, PDF file, 2.4 MB.

## ACKNOWLEDGMENTS

D.Y., S.N., Y.C., X.G., and J.W. collected the skin samples. M.L., W.S., X.L., J.B., H.J., and Y.C. collected the PM samples and performed part of the laboratory work. D.Y. and M.L. performed the laboratory work and wrote the manuscript. T.Z. performed part of the laboratory work and revised the manuscript. L.Y. and Z.T. designed the research and revised the manuscript. D.Y., M.L., M.W., X.S., and T.Z. analyzed the data for the work. All authors contributed significantly in the preparation of the manuscript. All authors approved of the submission of this manuscript.

This research was supported by the Science and Technology Research Project of Henan Province (no. 202102310270 and 212102310184), the CAMS Innovation Fund for Medical Sciences (CIFMS) (no. 2021-I2M-1-055), the National Microbial Resources Center (no. NMRC-2021-3), the National Natural Science Foundation of China (NSFC) (no. 81973220 and 32000006), the Non-profit Central Research Institute Fund of CAMS (no. 2020-PT310-003 and 2021-PT350-001), the Joint Construction Project of

Health Commission of Henan Province (no. LHGJ20190469), and the College Student Innovation and Entrepreneurship Training Program of Henan Province (no. 202110472018).

We declare no competing interests or conflict of interest.

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
