## [Reviewer comments · Microbiology Spectrum]

Microbiology Spectrum

Haze exposure changes the skin fungal community and promotes the growth of *Talaromyces* strains

Dong Yan, Min Li, Wenhao Si, Shijun Ni, Xin Liu, Yahan Chang, Xiaochan Guo, Jingjing Wang, Jie Bai, Yuanhang Chen, Haoyue Jia, Tao Zhang, Minna Wu, Xiangfeng Song, Zhongwei Tian, and Li-Yan Yu

Corresponding Author(s): Li-Yan Yu, Chinese Academy of Medical Sciences & Peking Union Medical College Institute of Medicinal Biotechnology

Review Timeline:

Submission Date:	April 1, 2022
Editorial Decision:	September 15, 2022
Revision Received:	October 7, 2022
Editorial Decision:	November 16, 2022
Revision Received:	November 16, 2022
Accepted:	November 18, 2022

Editor: Soo Chan Lee

Reviewer(s): Disclosure of reviewer identity is with reference to reviewer comments included in decision letter(s). The following individuals involved in review of your submission have agreed to reveal their identity: Laura Tipton (Reviewer #1); Fabio Palmieri (Reviewer #2)

Transaction Report:

DOI: <https://doi.org/10.1128/spectrum.01188-22>

September 15, 2022

Prof. Li-Yan Yu
Institute of Medicinal Biotechnology, Chinese Academy of Medical Sciences & Peking Union Medical College
TianTanXiLi 1, Dongcheng District
Beijing 100050
China

Re: Spectrum01188-22 (The haze exposure changes the skin fungal community and promotes the growth of *Talaromyces* strains)

Dear Prof. Li-Yan Yu:

Link Not Available

Sincerely,

Soo Chan Lee

Journals Department
Reviewer comments:

Reviewer #1 (Comments for the Author):

The authors have characterized the relationship between the human skin mycobiome and haze exposure, or particulate matter. They found that haze exposure does alter the composition and network properties of the mycobiome, and specifically increases abundance of *Talaromyces*. Increase in *Talaromyces* was confirmed by culturing isolates with PM.

While the science seems mostly fine, the manuscript was difficult to read. The following suggestions would help improve readability:

1. Review for grammar - there are a large number of errors, particularly with the use of "respectively" and "furtherly".

2. Reorder the results so that it goes diversity, differential abundance, networks, then culture. By starting with networks, I have little frame of reference for what differences are being analyzed. Similarly, I would recommend starting with the paragraph describing the taxa identified (starting on line 153) as this gives context for everything else.
3. It is not clear how many samples are in each group, in particular when building the networks. It looks like there are about 18 samples used for each network, which is not enough to draw strong conclusions about network topography. However, network construction is missing from the methods section and this could be addressed in that section.
4. The discussion section is missing some context or explanations beyond citations, particularly around cleaning the retroauricular crease (lines 260-261), airborne fungi (lines 269-277), and the neutral model (lines 314-319).

Smaller edits include:

1. Report all p-values as values rather than $>$ or $<$ 0.05.
2. Co-culture usually implies multiple microbes but it appears to mean one fungal strain with particulate matter in this case. For this situation, you can either use "culture" or define co-culture when it is first introduced.
3. In the culture section the abbreviation HC is used but not defined. HC, MC, and LC are all used in Fig 5 without definitions.

Reviewer #2 (Comments for the Author):

The authors describe here an interesting study where they assessed the influence of haze pollutants on the skin fungal communities depending on the host sex, as well as the seasons (spring vs. winter). The genus *Talaromyces* was found to be enriched during haze days. Moreover, the authors were able to isolate five *Talaromyces* strains and further showed that their growth was promoted by a high concentration of particulate matter.

There are a lot of parts in the manuscript that I could not properly understand. Comments and suggestions can be found in the review attachment file.

Staff Comments:

Preparing Revision Guidelines

Please return the manuscript within 60 days; if you cannot complete the modification within this time period, please contact me. If you do not wish to modify the manuscript and prefer to submit it to another journal, please notify me of your decision immediately so that the manuscript may be formally withdrawn from consideration by Microbiology Spectrum.

L1 : remove « the » in the title.
L24, 26, 27 & 30: remove “the”.
L31: what does “PD” stand for? Explain the acronym at the first instance.
L36: write “5” in full letters.
L39: replace “furtherly” by “furthermore”.
L40-41: “The genera significantly enriched...”, please rephrase.
L46: replace “on” by “in”.
L48-49: please rephrase.
L50: please define the acronym “PM” in line 46, e.g. Particular matter (PM).
L52: replace “thoughts” by “insights on the role”.
L52-54: split the sentence in two.
L61: add “the” before “host” and “skin”.
L62-64: please rephrase.
L65: I would suggest to replace “stay in temporal stability” by “is stable overtime”.
L66-68: please rephrase in order to avoid the repetition of “factors”.
L82-83: please rephrase the sentence “PM deposits ...”.
L91: please replace “to our best” by “to the best of our”.
L95: please rephrase “genera in strain-level”.
L100: please rephrase “... passed quality filtering, denoised, merged, chimeras filtering, ...”.
L103-107: please rephrase.
L113: what do you mean exactly by “classified”. Please clarify.
L140: please define “PD whole tree index”.
L165: remove “s” to “examples”.
L173: please define what a LEfSe analysis is.
L189: please rephrase “which was cluster to four groups”.
L193-194: please rephrase.
L195-196: please define the acronyms “LC”, “MC”, and “HC” at their first instances.
L204: please define “AIC”.
L220-339: please review thoroughly all the discussion and conclusion so that the reading is smoother.
L358-360: please write the date in the following format “on March 14” or “on the 14th of March”.
L373: please describe the PCR conditions that were used.
L375-377: How was the library preparation for the Illumina sequencing done?
L379: write “UNITE” in capital letters.
L394: “The swab sample was diluted 6 times...” in what was it diluted?
L395: what is the brand of the SDA? Is it also OXOID? If yes, please specify again.
L396: change from “To inhibit the bacteria growth” to “To inhibit bacterial growth”.
L401: again here, please indicate the PCR conditions and which sequencing technology was used.

Response to Reviewer #1's comments of the Manuscript "The haze exposure changes the skin fungal community and promotes the growth of *Talaromyces* strains" (Manuscript ID: Spectrum01188-22)

Reviewer #1 (Comments for the Author):

The authors have characterized the relationship between the human skin mycobiome and haze exposure, or particulate matter. They found that haze exposure does alter the composition and network properties of the mycobiome, and specifically increases abundance of *Talaromyces*. Increase in *Talaromyces* was confirmed by culturing isolates with PM.

Thanks for helping us to improve our manuscript. According to your comments, we revised the manuscript and uploaded our revisions and responses. The answers to the comments are as follows.

While the science seems mostly fine, the manuscript was difficult to read. The following suggestions would help improve readability:

1. Review for grammar - there are a large number of errors, particularly with the use of "respectively" and "furtherly".

Thank you for your suggestions. We checked the manuscript carefully and found an experienced professor to revise our manuscript for correcting the English. The incorrect uses of "respectively" were deleted in Line 117 and 192. The words "furtherly" were replaced with "furthermore" in Line 38, 160, and 320.

2. Reorder the results so that it goes diversity, differential abundance, networks, then culture. By starting with networks, I have little frame of reference for what differences are being analyzed. Similarly, I would recommend starting with the paragraph describing the taxa identified (starting on line 153) as this gives context for everything

else.

OK, we reorganized the results according to your suggestions. The contents of results go composition, diversity, differential abundance, networks, then culture.

3. It is not clear how many samples are in each group, in particular when building the networks. It looks like there are about 18 samples used for each network, which is not enough to draw strong conclusions about network topography. However, network construction is missing from the methods section and this could be addressed in that section.

Thank you for your suggestions, which remind us to improve our research. The samples numbers were listed in Line 97-100. And we revised the method of network to ‘19 samples (haze days in spring group), 19 samples (non-haze days in spring group), 18 samples (haze days in winter group), and 18 samples (non-haze days in winter group) were used to construct four networks (Fig 2A). The networks were constructed using the Cytoscape (version 3.8.2) based on Spearman correlation index (cutoff value > 0.7) (80,81). Using R software, the network structure was described with natural connectivity in response to node removal and degree distribution.’ in Line 384-390.

4. The discussion section is missing some context or explanations beyond citations, particularly around cleaning the retroauricular crease (lines 260-261), airborne fungi (lines 269-277), and the neutral model (lines 314-319).

(1) ‘cleaning the retroauricular crease (lines 260-261)’: We deleted the sentence ‘Secondly, the males may have low frequency of cleaning the retroauricular crease, which lead to the longer times to haze exposure.’, because no reference was found to support our speculation.

(2) 'airborne fungi (lines 269-277)': We revised to 'Alternaria and Cladosporium were the dominant fungal genera in the skin samples. Given that *Alternaria* and *Cladosporium* were the most prevalent genera in the air (61,62) and dominated the airborne fungal community in China (31), which indicated that these fungal genera on the skin might come from the air.' in Line 260-262.

(3) 'the neutral model (lines 314-319)': Sorry, the original version is unclear. We revised to 'In addition, taxa with significant differences between haze and non-haze days such as *Talaromyces* and Pezizaceae diverge from the neutral model, which suggests that they have stronger colonization potential (21, 72, 73).' In Line 310-313.

Smaller edits include:

1. Report all p-values as values rather than > or < 0.05.

We revise the p-values as values in Line 129 ($p = 0.028$), Line 129 ($p = 0.045$), Line 115 ($p = 0.42$), Line 116 ($p = 0.013$), and Line 117 ($p = 0.001$).

2. Co-culture usually implies multiple microbes but it appears to mean one fungal strain with particulate matter in this case. For this situation, you can either use "culture" or define co-culture when it is first introduced.

Thank you for your suggestions, we revised 'co-culture' to 'culture' in Line 36, 320, 330, 420, and 426.

3. In the culture section the abbreviation HC is used but not defined. HC, MC, and LC are all used in Fig 5 without definitions.

We added the definitions 'The spore suspension of *Talaromyces* strains were cultured with PM at 0 mg/mL (Control), 0.08 mg/mL (LC group), 0.64 mg/mL

(MC group), 5.12 mg/mL (HC group) for 5 days.’ in Line 194-196, which we also defined in Fig 6.

Response to Reviewer #2’s comments of the Manuscript “The haze exposure changes the skin fungal community and promotes the growth of *Talaromyces* strains” (Manuscript ID: Spectrum01188-22)

Reviewer #2 (Comments for the Author):

The authors describe here an interesting study where they assessed the influence of haze pollutants on the skin fungal communities depending on the host sex, as well as the seasons (spring vs. winter). The genus *Talaromyces* was found to be enriched during haze days. Moreover, the authors were able to isolate five *Talaromyces* strains and further showed that their growth was promoted by a high concentration of particulate matter.

Thanks for your suggestions. According to your and other reviewer’s comments, we have revised our paper carefully. Thank you for helping us to improve the manuscript.

There are a lot of parts in the manuscript that I could not properly understand.

Comments and suggestions can be found in the review attachment file.

L1 : remove « the » in the title.

We removed “the” in the title.

L24, 26, 27 & 30: remove “the”.

We revised ‘The haze pollutant’ to ‘haze pollutant’ in Line 24, ‘the human health’ to ‘human health’ in Line 26, ‘The significant’ to ‘Significant’ in Line 27, ‘The fungal network’ to ‘Fungal network’ in Line 35.

L31: what does “PD” stand for? Explain the acronym at the first instance.

We explained the ‘PD’ as ‘Phylogenetic diversity’ at the first instance in Line 30 and 123.

L36: write “5” in full letters.

OK, we revised “5” to “five” in Line 36.

L39: replace “furtherly” by “furthermore”.

We replaced “furtherly” by “furthermore” in Line 38.

L40-41: “The genera significantly enriched...”, please rephrase.

We rephrased to “*Talaromyces* enriched during haze days deviated from the neutral assembly process.” in Line 40-41.

L46: replace “on” by “in”.

We replaced “on” by “in” in Line 45.

L48-49: please rephrase.

We revised to “haze exposure influenced the diversity, composition, and network of skin fungal community” in Line 47-48.

L50: please define the acronym “PM” in line 46, e.g. Particular matter (PM).

Thank you for your suggestion. We defined as Particular matter (PM) in Line 45.

L52: replace “thoughts” by “insights on the role”.

We replaced “thoughts” by “insights on the role” in Line 51.

L52-54: split the sentence in two.

We split the sentence to “We anticipate that this study may provide new insights on the role of haze exposure disturbing the skin fungal community. It lays the groundwork for further clarifying the association between the changes of skin fungal community and adverse health outcomes.” in Line 51-54.

L61: add “the” before “host” and “skin”.

We added “the” before “host” and “skin microbiota” in Line 60-61.

L62-64: please rephrase.

We reshaped to “Fungi are less abundant than bacteria on the skin (4), but they play important roles in skin health (5, 6). Fungal communities are ignored by most of the studies on skin microbiota using high-throughput sequencing (7-10).” in Line 62-64.

L65: I would suggest to replace “stay in temporal stability” by “is stable overtime”.

We replaced “stay in temporal stability” by “is stable overtime” in Line 65.

L66-68: please rephrase in order to avoid the repetition of “factors”.

We replaced “environmental factors” to “environmental variables” in Line 66.

L82-83: please rephrase the sentence “PM deposits ...”.

We rephrased to “PM deposits on the skin and constitutes a part of the skin microbiota. The pollutants attached in PM may disturb the skin microbiota, affecting skin health.” in Line 81-83.

L91: please replace “to our best” by “to the best of our”.

We replaced “to our best” by “to the best of our” in Line 90.

L95: please rephrase “genera in strain-level”.

We replaced “genera in strain-level” to “representative strains of the genera” in Line 94.

L100: please rephrase “... passed quality filtering, denoised, merged, chimeras filtering, ...”.

We rephrased “... passed quality filtering, denoised, merged, chimeras filtering, ...” to “...were quality filtered, denoised, merged, chimeras filtered, and clustered into...” in Line 105-106.

L103-107: please rephrase.

We rephrased to “which suggested that the following analysis should be classified to spring group and winter group.” and deleted the following sentence “To investigate whether the haze exposure influence the composition of the skin fungal community, the PM samples were classified to samples in spring and in winter.” in Line 109-110.

L113: what do you mean exactly by “classified”. Please clarify.

We reshaped to “we grouped the samples into female and male groups.” in Line 117-118.

L140: please define “PD whole tree index”.

We defined “PD whole tree index” to “Phylogenetic diversity (PD) whole tree index” in Line 123-124.

L165: remove “s” to “examples”.

We removed “s” to “examples” in Line 147.

L173: please define what a LEfSe analysis is.

We defined “LEfSe analysis” to “Linear discriminant analysis effect size (LEfSe)” in Line 154.

L189: please rephrase “which was cluster to four groups”.

We revised to “which clustered into four groups” in Line 190.

L193-194: please rephrase.

We rephrased to “XSF1, XSF7, XSF10, and XSF103 were selected to investigate the influence of PM on *Talaromyces* strains *in vitro*.” in Line 192-194.

L195-196: please define the acronyms “LC”, “MC”, and “HC” at their first instances.

We added the definitions ‘The spore suspension of *Talaromyces* strains were cultured with PM at 0 mg/mL (Control), 0.08 mg/mL (LC group), 0.64 mg/mL (MC group), 5.12 mg/mL (HC group) for five days.’ in Line 194-196.

L204: please define “AIC”.

We defined Akaike information criterion as AIC in Line 205.

L220-339: please review thoroughly all the discussion and conclusion so that the reading is smoother.

Thank you for your suggestion. We reviewed the discussion and conclusion and made revisions.

L358-360: please write the date in the following format “on March 14” or “on the 14th of March”.

We revised as “on March 14” in Line 350-352.

L373: please describe the PCR conditions that were used.

We described the PCR conditions in Line 363-369.

L375-377: How was the library preparation for the Illumina sequencing done?

We described the library preparation for the Illumina sequencing in Line 370-372.

L379: write “UNITE” in capital letters.

We revised UNITE in capital letters in Line 376.

L394: “The swab sample was diluted 6 times...” in what was it diluted?

The head of the swab was inserted into the tube containing the solution which we described in Line 343-344. Therefore, we revised “The swab sample was diluted” to “The buffer-containing swab sample was diluted” in Line 391-392.

L395: what is the brand of the SDA? Is it also OXOID? If yes, please specify again.

We specified as OXOID in Line 394.

L396: change from “To inhibit the bacteria growth” to “To inhibit bacterial growth”.

OK, we changed from “To inhibit the bacteria growth” to “To inhibit bacterial growth” in Line 394-395.

L401: again here, please indicate the PCR conditions and which sequencing technology was used.

We indicated the PCR conditions and sequencing technology in Line 398-402.

November 16, 2022

Prof. Li-Yan Yu
Institute of Medicinal Biotechnology, Chinese Academy of Medical Sciences & Peking Union Medical College
TianTanXiLi 1, Dongcheng District
Beijing 100050
China

Re: Spectrum01188-22R1 (Haze exposure changes the skin fungal community and promotes the growth of *Talaromyces* strains)

Dear Prof. Li-Yan Yu:

Thank you for submitting your manuscript to Microbiology Spectrum. Your manuscript is provisionally accepted. Please see the reviewer 1's comments and provided appropriate response. When submitting the revised version of your paper, please provide (1) point-by-point responses to the issues raised by the reviewers as file type "Response to Reviewers," not in your cover letter, and (2) a PDF file that indicates the changes from the original submission (by highlighting or underlining the changes) as file type "Marked Up Manuscript - For Review Only". Please use this link to submit your revised manuscript - we strongly recommend that you submit your paper within the next 60 days or reach out to me. Detailed instructions on submitting your revised paper are below.

Link Not Available

Sincerely,

Soo Chan Lee

Journals Department
Reviewer comments:

Reviewer #1 (Comments for the Author):

Thank you for addressing our concerns in this revision, the manuscript is much improved. I appreciate the repeated reminders of the sample size. However, given the small sample size and large number of ASVs, I feel that the network analysis is inappropriate and should be removed. Spearman correlation is not appropriate for abundance data without a transformation (many references exist about this in the bacteria literature) and the small sample size limits any conclusions that can be drawn about a network. That said, the paper is strong without this analysis.

Reviewer #2 (Comments for the Author):

Thank you to the authors for their work on the manuscript, which quality is improved thanks to the revisions. I do not have any comment to add.

Staff Comments:

Preparing Revision Guidelines

Please return the manuscript within 60 days; if you cannot complete the modification within this time period, please contact me. If you do not wish to modify the manuscript and prefer to submit it to another journal, please notify me of your decision immediately so that the manuscript may be formally withdrawn from consideration by Microbiology Spectrum.

Dear Editor,

Thank you for your review work. We have revised the manuscript according to Reviewer #1's comments. The response is as follows

Response to Reviewer #1's comments of the Manuscript "The haze exposure changes the skin fungal community and promotes the growth of *Talaromyces* strains" (Manuscript ID: Spectrum01188-22)

Reviewer comments:

Reviewer #1 (Comments for the Author):

Thank you for addressing our concerns in this revision, the manuscript is much improved. I appreciate the repeated reminders of the sample size. However, given the small sample size and large number of ASVs, I feel that the network analysis is inappropriate and should be removed. Spearman correlation is not appropriate for abundance data without a transformation (many references exist about this in the bacteria literature) and the small sample size limits any conclusions that can be drawn about a network. That said, the paper is strong without this analysis.

Thanks for your suggestions. According to your comments, we have revised our paper carefully.

We removed all the sentences related to the network analysis in Abstract (Line 35), Importance (Line 45), Keywords (Line 57-58), Introduction (Line 92), Results (166-185), Discussion (Line 224, 293-301), Conclusion (Line 320), Materials and methods (Line 386-392), and Fig 5 (Line 739-747) in the Marked-up PDF version.

November 18, 2022

Prof. Li-Yan Yu
Chinese Academy of Medical Sciences & Peking Union Medical College Institute of Medicinal Biotechnology
TianTanXiLi 1, Dongcheng District
Beijing 100050
China

Re: Spectrum01188-22R2 (Haze exposure changes the skin fungal community and promotes the growth of *Talaromyces* strains)

Dear Prof. Li-Yan Yu:

Your manuscript has been accepted, and I am forwarding it to the ASM Journals Department for publication. You will be notified when your proofs are ready to be viewed.

Sincerely,

Soo Chan Lee
Editor, Microbiology Spectrum
